# Zero-shot Gait Classification with Diffusion Models

**Xiaodong Guan**                                         XIAODONG.GUAN.21@UCL.AC.UK
**Robert Gray**                                                    R.GRAY@UCL.AC.UK
**Ashwani Jha**                                              ASHWANI.JHA@UCL.AC.UK
**Parashkev Nachev**                                            P.NACHEV@UCL.AC.UK
*Queen Square Institute of Neurology, University College London*

**Editors:** Accepted for publication at MIDL 2025

## Abstract

Movement disorders such as Parkinson's disease are characterised by complex abnormalities of body motion that resist precise, replicable, and scalable quantification. Subjective clinical scores–the established standard–are limited in expressivity and vulnerable to intra-observer variation; wearable sensor-based methods offer objectivity but with limited anatomical sampling. Remote video-based approaches could deliver both highly expressive and objective quantification of motion, but sufficient labelled samples are hard to obtain under clinical data regimes. Here we develop a diffusion model-based, zero-shot, and human-interpretable approach to gait assessment from video-derived pose data and evaluate it in Parkinson's Disease. Capable of detecting subtle changes in body motion without explicit training, it shows potential for an accurate, robust, and scalable solution, addressing the major limitations of existing methods.

**Keywords:** Gait Analysis, Generative Models, Diffusion Models.

## 1. Introduction

Movement disorders such as Parkinson's Disease (PD) involve complex abnormalities of body motion that are critical to diagnosis, monitoring, and treatment selection. Gait is a key aspect here, since it is both frequently affected and of great functional significance. Clinicians typically use the Movement Disorder Society-sponsored revision of the Unified Parkinson's Disease Rating Scale (MDS-UPDRS) (Goetz et al., 2008), which relies on observational judgments. However, their subjective nature poses challenges in reproducibility and precise quantification.

Emerging machine-learning approaches quantify gait parameters through computer-vision methods, analyzing features such as feet distance(Verlekar et al., 2018), swing velocity(Eltoukhy et al., 2017), 2D body pose(Rupprechter et al., 2021; Jinila et al., 2022; Tan et al., 2024; Nõmm et al., 2016), gait energy images(Ortells et al., 2018), and body keypoints in Cartesian space(Kaur et al., 2022); or through wearable-device-based feature extraction(Han et al., 2023; Moreau et al., 2023). Although promisingly performant on selected test datasets, these methods have limitations such as reliance on specially designed environments, customized camera setups, tailored sensors, and abundant labelled data. Moreover, most measurements are conducted in Cartesian or 2D space, lacking translation- and appearance-invariance, thus contributing to poor generalizability.

To address these limitations, here we propose a zero-shot diffusion model pretrained on diverse human motions, quantifying the deviation of input actions from specific textual

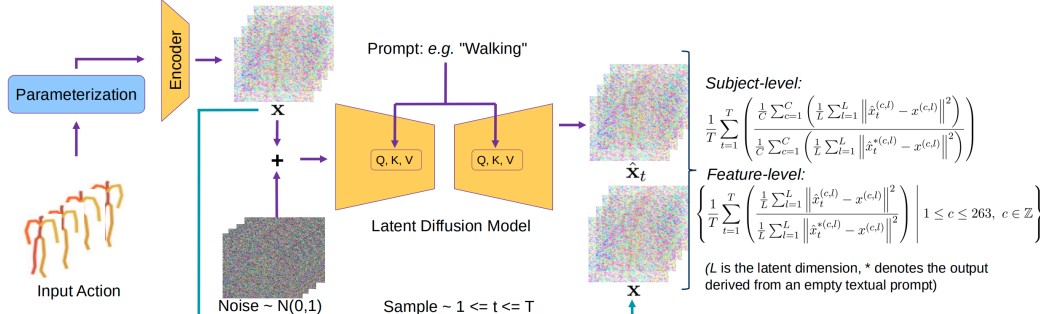

Figure 1: Proposed framework for quantifying the deviation of a test action from its prediction under a given prompt, yielding a measure of movement anomaly.

action descriptions. The approach offers a high-fidelity mapping between textual descriptions and 3D pose sequences, allowing for scale-, translation-, and viewpoint-invariant gait representations. At test time, the model evaluates gait clips against generated motions, producing a robust anomaly score that aligns well with clinician assessments, without requiring task-specific training or annotated datasets. Evaluated in PD, this approach enables sensitive, scalable, and automated gait assessment for early disease detection in clinical and real-world settings, and is transferable to other clinical scenarios.

## 2. Method

Videos of patients walking were collected during an ethically approved prospective dual-site medical device evaluation study (Jha et al., 2020). We parameterized body movements over 1-1.6 second intervals, indexed by 24 video-derived joint positions sampled at 25Hz, using axis-angle representation, and converted them into the HumanML3D format (Guo et al., 2022), neutralizing the effects of body shape. A latent diffusion model (Tevet et al., 2022) was used to model the distribution of movements, conditioned on their textual descriptions, providing an index of the anomaly of test movements qualified under a normative description(Li et al., 2023). Instead of evaluating predicted noise(Li et al., 2023), we compared predicted vs original latent features at each sampled time-step, formulating a subject-level anomaly measure as the mean squared error across all latent dimensions and feature channels, and a feature-level anomaly measure as the mean squared error across all latent dimensions. Both measures are normalized using values derived from an empty textual prompt, then averaged over time (see Figure 1).

## 3. Experiments and Discussion

**Result** We evaluated the proposed framework on gait data from 62 patients, divided into 4 groups based on MDS-UPDRS Part 3 gait scores assessed within normal medical treatment regime (0–3 in this cohort, where 0 indicates normal gait and higher scores reflect progressively worsening motor function), using 400 gait clips (100 per group) recorded with various devices, including smartphones and webcams. The evaluation focused on the correspondence between our anomaly measure and MDS-UPDRS scores. For both subject- and feature-level measures, we used the first 900 of the 1000 total time-steps, as they provided the greatest distinction between prompts. Using the prompt "Walking", the subject-level measure yielded a Spearman correlation coefficient of 0.7011 ($p <$1e-5). A one-

way ANOVA comparing between- and within-group variance produced an F-value of 186.14 ($p <$1e-5). The significance threshold for both tests was 0.05. The parameterized motion data consisted of 263 dimensions, capturing interpretable features including Cartesian joint positions, joint rotations (as 6D matrices(Zhou et al., 2019)), and linear velocities, for which feature-level anomaly measures are derived. In Figure 2, the first three panels show comparisons of each non-zero MDS-UPDRS score level against normal (MDS-UPDRS=0) gait, showing that the gaits of MDS-UPDRS $> 0$ groups exhibit greater errors than those from the MDS-UPDRS=0 group, with concentrations in the lower limbs and distal joints such as the feet and hands. Joint position and rotation errors were also prominently affected in the head and knee regions. Velocity-based errors were larger at the hip and spine, likely due to rigidity causing greater deviation from typical gait patterns. The right-most panel shows the correlation between feature-level anomaly measures and MDS-UPDRS scores, further highlighting that shoulder, lower limb, and head regions are most strongly associated with MDS-UPDRS when measuring Cartesian position; distal joints (hands, head, feet) dominate for rotations; and the pelvic region contributes most in the velocity domain. These results suggest that reduced dexterity, flexibility, and impaired joint coordination are key contributors to gait abnormalities in PD. The anatomical distribution of error also indicates imbalanced movement, consistent with PD-related asymmetry.

**Conclusion** We introduce an interpretable diffusion model-based gait classification framework for quantifying abnormalities of gait that combines expressivity with objectivity, and is operable within the few-label data regimes clinical scenarios impose. In the context of PD, our zero-shot approach enables the quantification of abnormalities–both feature- and subject-level–with good fidelity without explicit training, facilitating implementation, and broadening access to automated analysis. The high correlation with MDS-UPDRS scores–obtained zero-shot–leaves room for substantial benefit from fine-tuning as larger volumes of data become available. Conditioning on highly expressive text labels enables ready extension of the framework to richly defined classification tasks, with utility across all clinical scenarios, across movement disorders and beyond, where quantification of body motion is critical to clinical management.

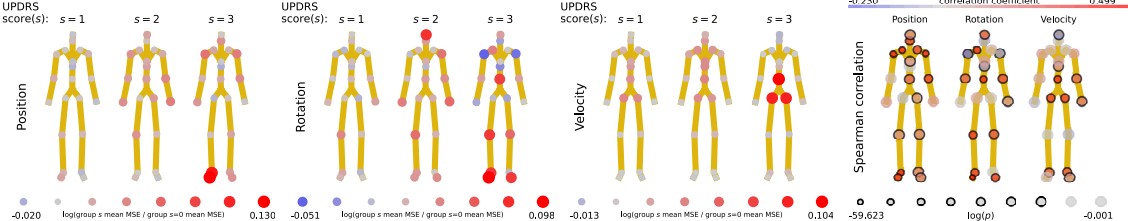

Figure 2: Anatomical projection of joint-wise error magnitudes across groups, along with the correlation strength contributed by errors at each joint. Outlined circles indicate regions below the corrected statistical significance threshold ($p = 0.0166$), with correlations computed independently for each data component.

## Acknowledgments

This study was funded by Wellcome Trust and supported by the National Institute for Health and Care Research University College London Hospitals Biomedical Research Center.

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
