# OpenReview forum: "Zero-shot Gait Classification with Diffusion Models"
_MIDL.io/2025/Short_Papers — MIDL 2025 - Short Papers_

### Official Review · Reviewer_Zduw · 2025-04-16

**Rating:** 3
**Confidence:** 2

**Summary:**

This paper proposes a zero-shot gait assessment framework using a latent diffusion model conditioned on textual prompts to quantify motor abnormalities in movement disorders, with a focus on Parkinson's disease. The method computes an anomaly score by comparing observed gait to the model’s expectation of "normal" motion, using 3D pose sequences derived from video data. It does not require disease-specific training data or labels. Evaluation on 62 PD patients shows strong correlation (p = 0.70) with MDS-UPDRS gait scores, demonstrating the model’s capacity to detect clinically relevant gait anomalies without supervision.

**Strengths:**

- Uses diffusion models in a zero-shot setting for clinical gait analysis, circumventing the need for large annotated datasets.
- Anatomical and feature-level visualizations provide insight into which body regions contribute most to the anomaly scores, aiding clinical trust.
- The use of text prompts and normalized latent comparisons suggests a framework extensible across various movement disorders.
- The Spearman correlation with MDS-UPDRS scores is high for a zero-shot method (ρ = 0.70), validating its potential.
- Can be applied across different devices (smartphones, webcams), enabling deployment in real-world, low-resource environments.

**Weaknesses:**

- While 400 clips were analyzed, only 62 patients were included. Larger cohorts would strengthen generalizability claims.
- Some aspects, such as pose estimation pipeline, clip preprocessing, or exact prompt formulation, are not deeply elaborated.
- The paper lacks benchmarking against existing gait analysis methods (e.g., pose-based classifiers, wearable-based scoring).
- Although some interpretability is offered, the inner workings of the diffusion prediction and prompt alignment process could be further clarified.

---

### Decision · Program_Chairs · 2025-05-01

Accept